# Synchronising Spatial Metadata Records and Interfaces to Improve the Usability of Metadata Systems

Mohsen Kalantari [1,*], Syahrudin Syahrudin [2] , Abbas Rajabifard [1] and Hannah Hubbard [1]

1   Department of Infrastructure Engineering, The University of Melbourne, Melbourne, VIC 3010, Australia;
    abbas.r@unimelb.edu.au (A.R.); hubbardh@unimelb.edu.au (H.H.)
2   Indonesian Geospatial Agency, Jakarta 16911, Indonesia; syahrudin@big.go.id
*   Correspondence: mohsen.kalantari@unimelb.edu.au

**Abstract:** The spatial data infrastructure literature reveals significant gaps in metadata systems concerning their efficiency and effectiveness for end-users. The literature proposes improvements to make the metadata systems more user-friendly. These improvements include new metadata elements and user interfaces that are in concert with each other. In this paper, we implement the proposed improvements in a prototype system and engage with end-users to assess if the proposals help users' expectations. The prototype is evaluated by conducting think-aloud protocol (TAP) usability testing and semi-structured interviews with end-users. The result demonstrates an increased level of satisfaction about existing systems and some more areas to improve. We conclude that a synchronised development approach for the spatial metadata and the user interface will increase the usability of the metadata for data discovery and selection.

**Keywords:** spatial data infrastructure (SDI); spatial metadata; user-centred design (UCD); usability

## 1. Introduction

Spatial data custodians design and develop metadata systems, including standards and user interfaces, to manage the data and make its sharing, discovery and use possible for end-users. The design of these systems often has minimal or no inputs from the users. The usability of the metadata systems is constantly scrutinised in the literature [1–3]. To this end, research studies have addressed usability issues from various perspectives.

A substantial body of research covers issues related to creating complete and consistent metadata records [4–8]. The outcomes of this research offer automatic approaches to creating and updating metadata records. There is also existing research on the involvement of users in metadata creation and on improving the discovery of spatial data using semantics and ontologies [9–22]. These papers take state-of-the-art approaches to the next level. They argue that the role of users in the creation, organisation and even discovery of metadata records should be augmented. More recently, research into user-centric spatial metadata systems has identified problems with the user experience [23] and metadata standards [24].

Based on the user-centred design (UCD) methodology [25], previous studies have engaged end-users to understand their view on the efficiency and effectiveness of existing metadata systems. Previous authors have [23] discovered significant gaps and usability issues in terms of spatial data discovery and selection. Building on this, [24] engaged 61 users of various existing data portals to delve deeper into the usability issues. The authors of [24] offered improved design principles for the interface in metadata systems based on the engagement result. This study [24] also provided an extension to the ISO 19115 standard to assist user needs for spatial data discovery and selection.

This paper aims to gauge users' acceptance of the newly proposed spatial metadata profile and user interface by [24]. This paper first analyses the interface requirements

concerning the extension. It then implements them in a prototype system. Finally, it evaluates the proposal by engaging end-users.

The following section presents the methodology, including developments of the prototype. Section 3 discusses the results and findings. The users' acceptance of the prototype, the usability of and reactions to the spatial metadata and the user-interface prototype are analysed and interpreted. Sections 4 and 5, respectively, discuss and conclude by providing a proposed solution for improving metadata usability for spatial data discovery and selection.

## 2. Methodology

The need for improving the usability of spatial metadata for spatial data users necessitates an understanding of the interaction between data users and the metadata. Previous authors [23] collected and analysed behavioural data of spatial data users. They used user-centred design (UCD) to understand the usability problems of metadata systems. We implemented a TAP, interviews and surveys as empirical UCD methods to observe the interaction of users with metadata systems, evaluate usability, identify usability problems and gather users' needs and expectations.

The authors of [24] take the usability issues discovered by [23] and developed a user-oriented spatial metadata profile to help achieve user-friendly metadata systems. The user-oriented profile extends ISO 19115 and adds elements that potentially facilitate data discovery and selection. These authors [24] also suggested some improvements in user interfaces used in metadata systems.

This paper builds on [23,24] and follows the UCD methodology [25] (Figure 1). It first creates a prototype system that implements a user-oriented spatial metadata profile and interfaces. Then it validates if the profile facilitates a better experience for discovery and selection by engaging users via the prototype system.

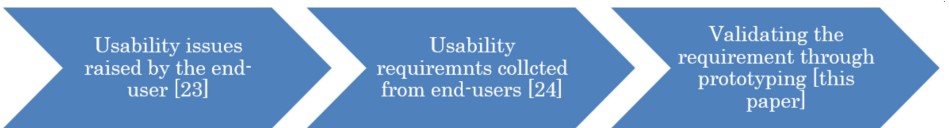

**Figure 1.** This paper is built on the existing literature.

The prototyping process begins with analysing the relationships between the spatial metadata elements from the profile proposed and the user-interface functionalities discovered by [24]. The purpose is to ensure all functionalities are in concert with respective metadata elements and vice versa. Here, a functional prototype spatial metadata system is created by providing the required information for the metadata elements of the selected established spatial data followed by putting them into a dedicated database. The user interfaces of the prototype are developed by designing a spatial data catalogue web page comprising identified functionalities and interfaces to support spatial data discovery and selection. The authors made an initial internal evaluation to ensure all functionalities are covered and functional, e.g., tabs, buttons, windows, and then used the reviews to refine the system. Once the prototype is ready, the final step of this validation stage is evaluating the prototype. The evaluation was conducted using a TAP by interviewing selected spatial data users. These users used the web catalogue prototype to discover and choose specific spatial data to be used in a project featuring a simulated real-life scenario (Figure 2). The following sections describe each step of the research in detail.

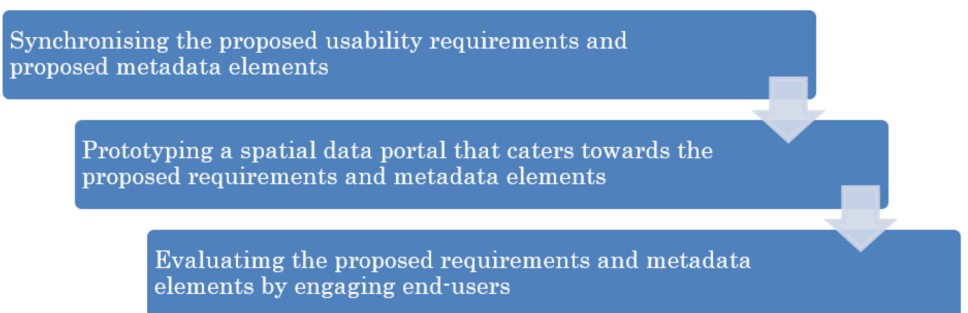

**Figure 2.** The methodology of the paper.

## 2.1. Synchronising

The process started by identifying the prototype's system functionalities based on the proposed profile by [24]. Besides the profile, user expectations and experience as described in [23] informed the functionalities and the presentation design of the interface. The result is a set of user-interface functionalities and the relationships between them and the metadata elements, as illustrated in Figure 3.

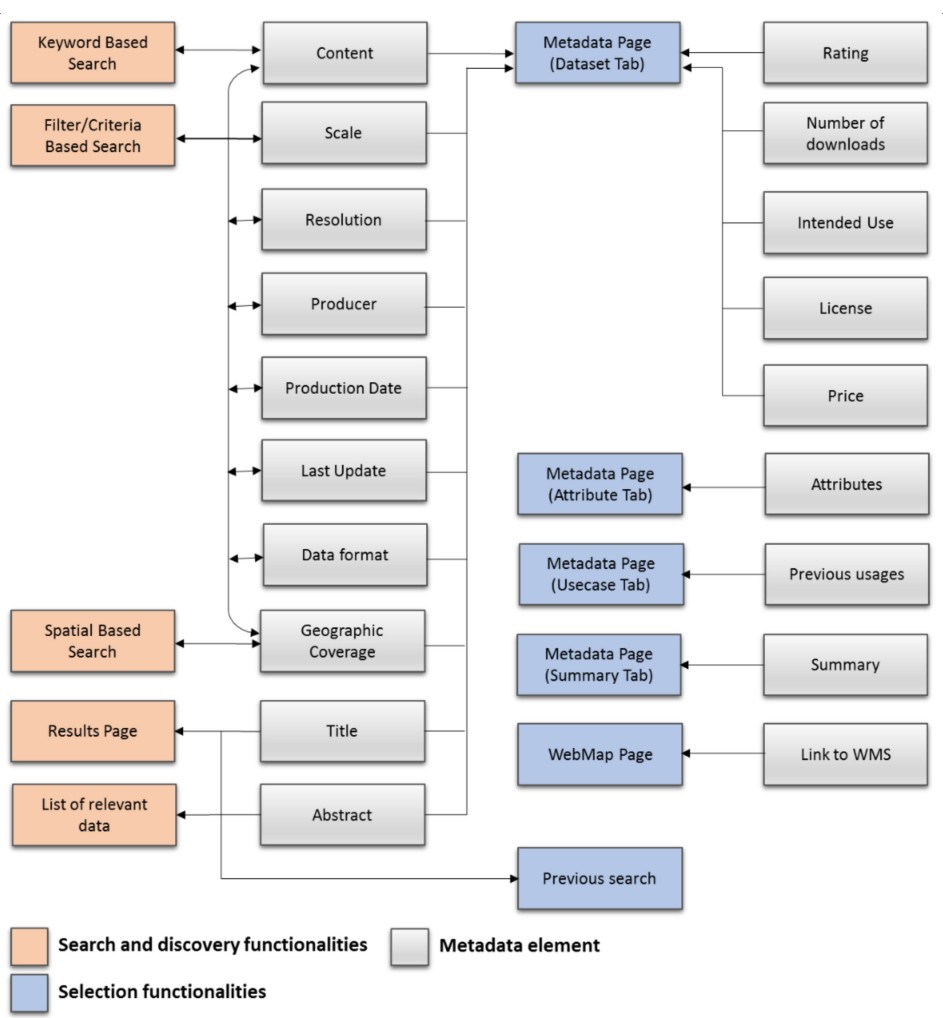

**Figure 3.** The relationship of metadata elements and user-interface functionalities.

The functionalities of the prototype are divided into two groups by [23]: Search and discovery functionalities and selection functionalities. The search and discovery functionalities are designed to help users find and discover potential spatial data for their projects. The selection functionalities are designed for users to select data based on their



suitability to the users' projects. Each functionality is related to one or more metadata elements in [24] as mapped in Figure 1. This mapping ensures that the metadata elements and user requirements are consistent and in concert before the prototype development starts.

### 2.2. Prototyping

The spatial data available from the Indonesian Spatial Data Infrastructure was selected for the purpose of prototyping. The selection of the spatial data is based on two criteria: Open access to spatial data and its services and a variety of themes. The prototype is implemented as a web portal that selected spatial data users can use for evaluation. The prototype consists of four main pages: The search page (as the homepage), the search results page covering the data search and discovery functionalities, the dataset metadata page (with tabs) and the web map service page covering data selection functionalities.

#### 2.2.1. Search Page

The search page is also the home page of the website and consists of three different search tools: (1) Keyword-based search, (2) criteria (filter)-based search and (3) spatial or location-based search, as illustrated in Figure 4. Users might use a search tool based on their knowledge and experiences. The keyword-based search is a simple search tool that users can use to type in specific words relevant to the spatial data they are interested in. The criteria-based search provides a set of criteria users can use to narrow down their search. The location-based search allows users to define their area of interest by drawing a boundary polygon on the map window before submitting their inquiry.

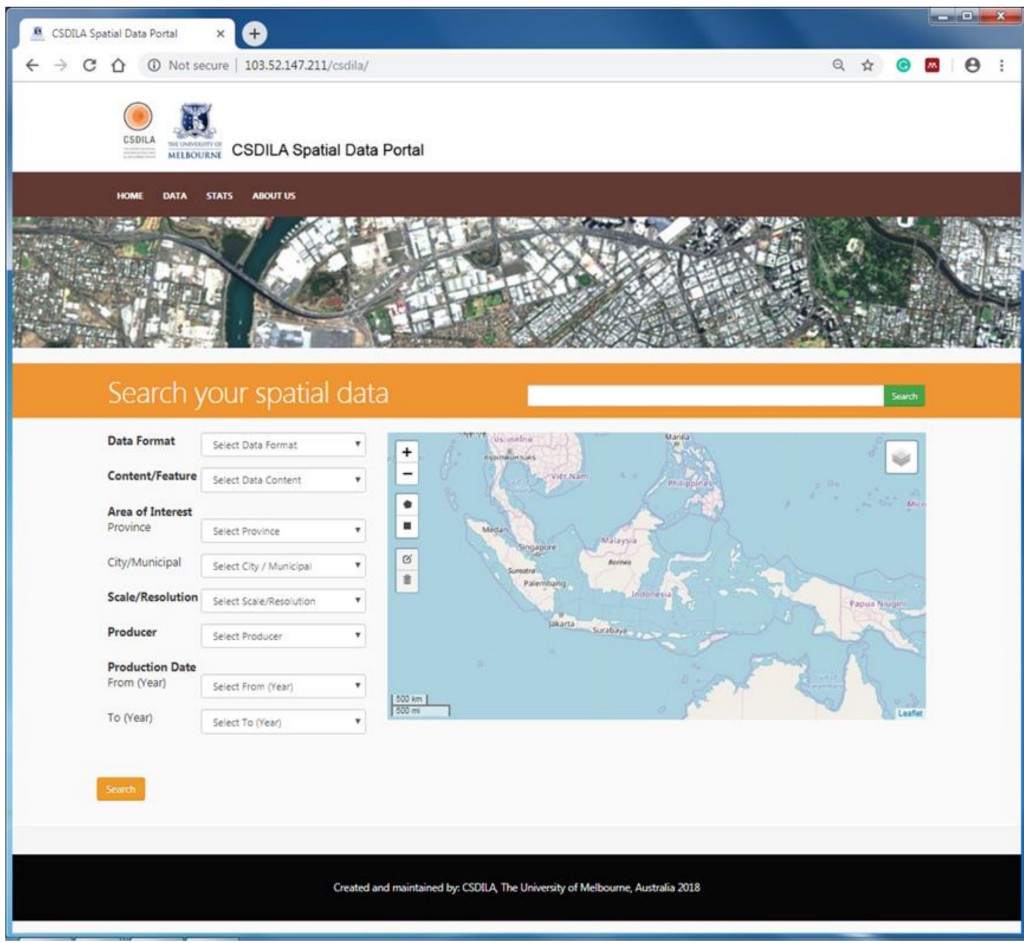

**Figure 4.** Search page (home page).

### 2.2.2. Search Results Page

The search results page consists of the main window and the relevant data window, as illustrated in Figure 5. The main window provides a list of data titles and abstracts responding to their submitted search. The relevant data window presents a list of data titles and abstracts related to the submitted search.

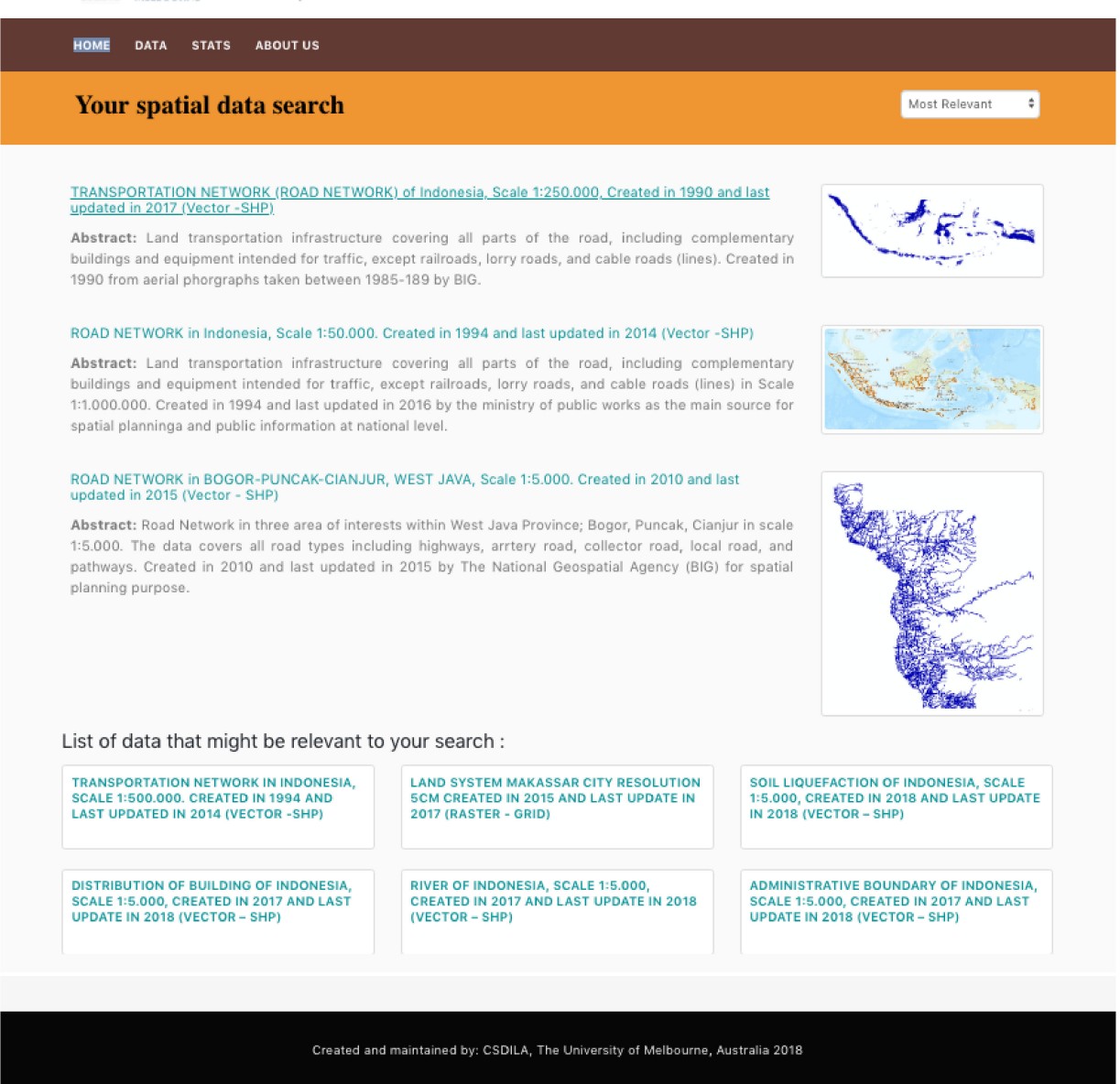

**Figure 5.** Results page.

### 2.2.3. Dataset Metadata Page

The dataset tab is the default metadata page presented to users when they click one of the titles on the results page. As illustrated in Figure 6, it consists of the primary metadata window showing data characteristics, such as the following user-related information: A data rating, the number of downloads, geographic coverage in a graphical form (thumbnail) and the previous search window providing the user the results (a list of data titles and abstracts) of their previous searches.

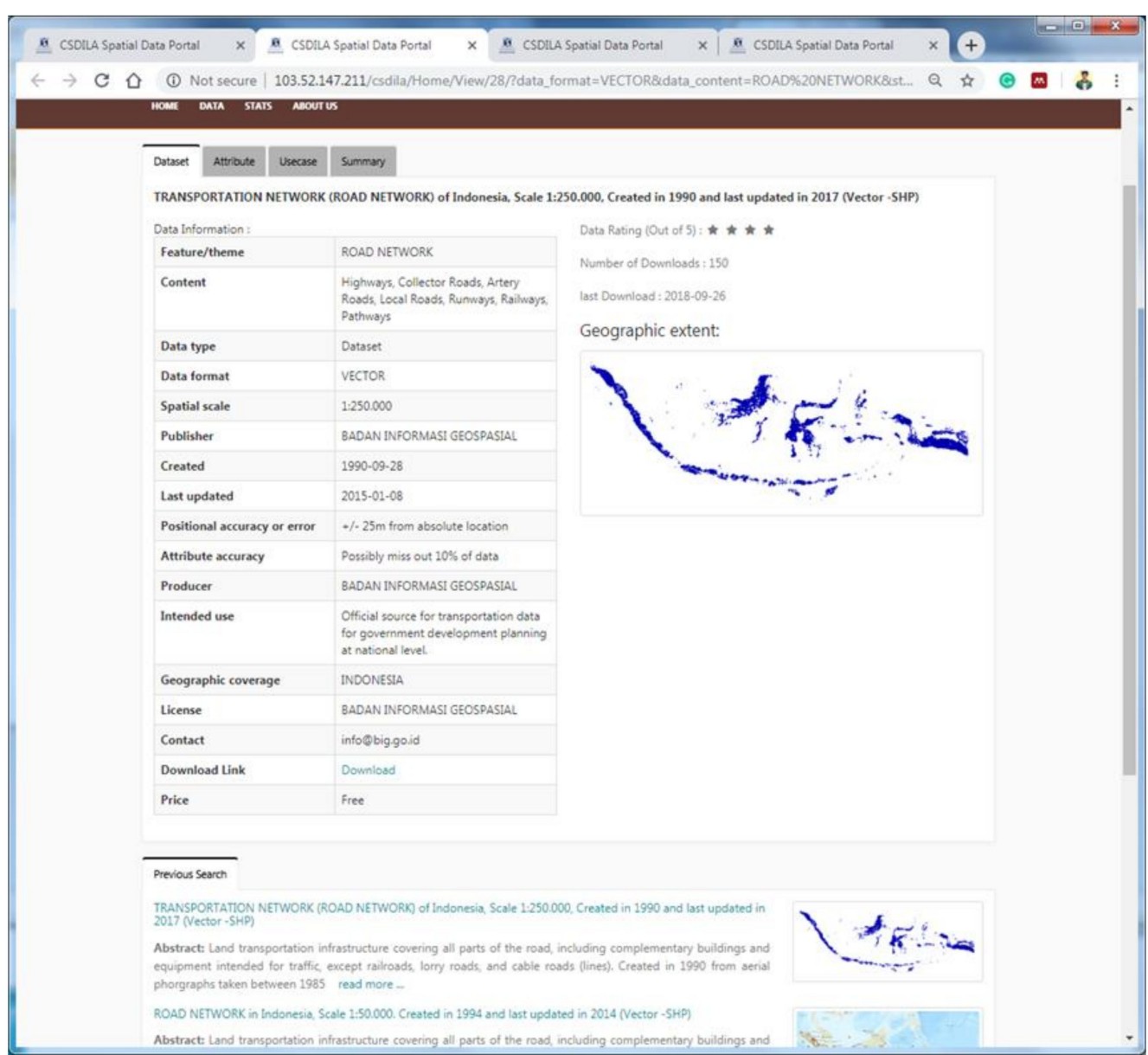

**Figure 6.** Metadata page (Dataset tab).

### 2.2.4. Attribute Metadata Page

The next tab provided as part of the metadata page is the attribute tab. It contains information about the spatial data attributes in a tabular format that might assist users in getting to know the contents and structure of the data. A previous search window is also provided below the main attribute window similar to the dataset tab, as illustrated in Figure 7.

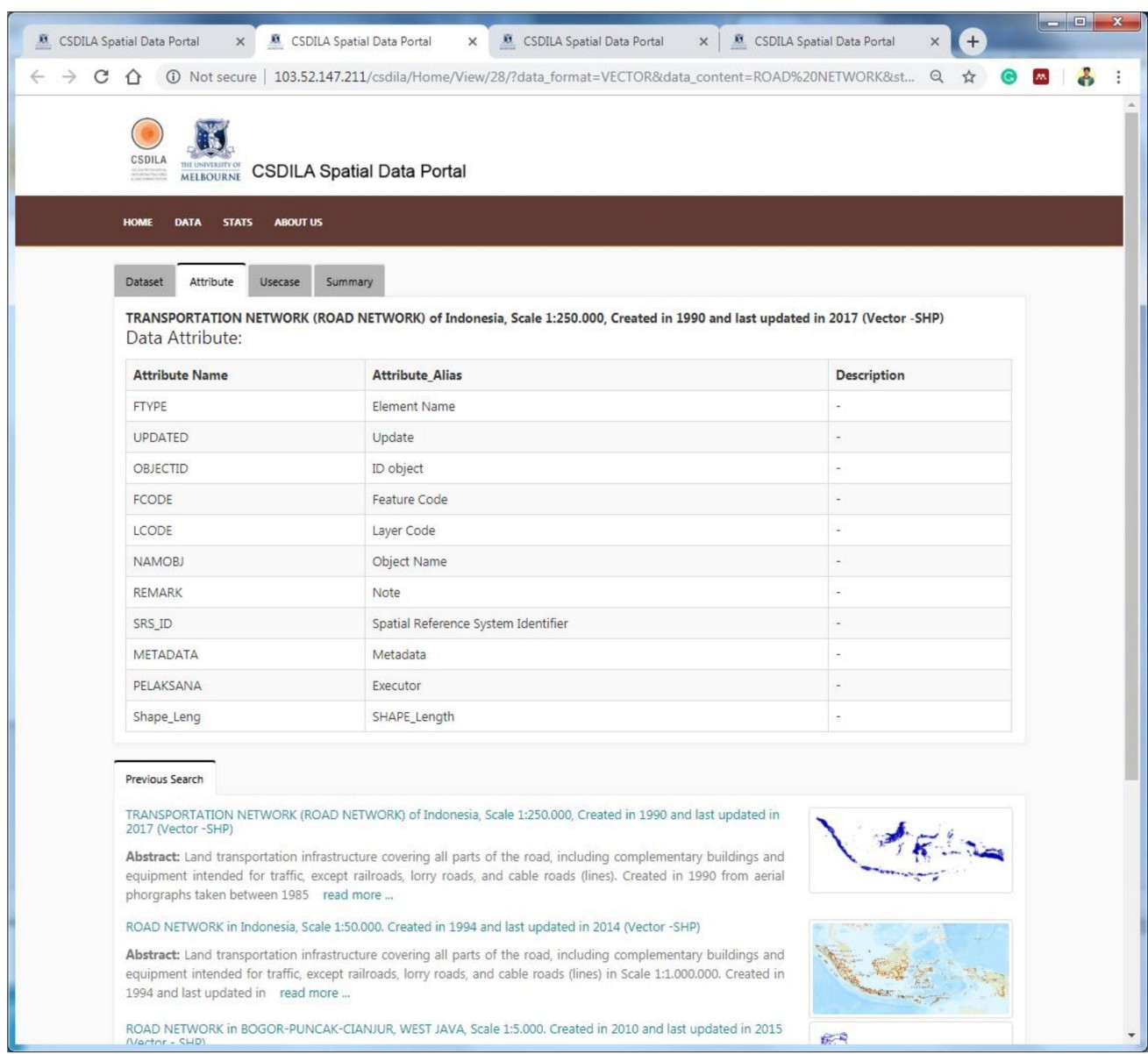

**Figure 7.** Metadata page (Attribute tab).

### 2.2.5. Usecase Metadata Page

Figure 8 illustrates the usecase metadata page, a tab that provides users with information about the previous usages of the data. In this tab, users can give their own experiences with the data by submitting reviews into a form. The previous search window is also provided on this page.

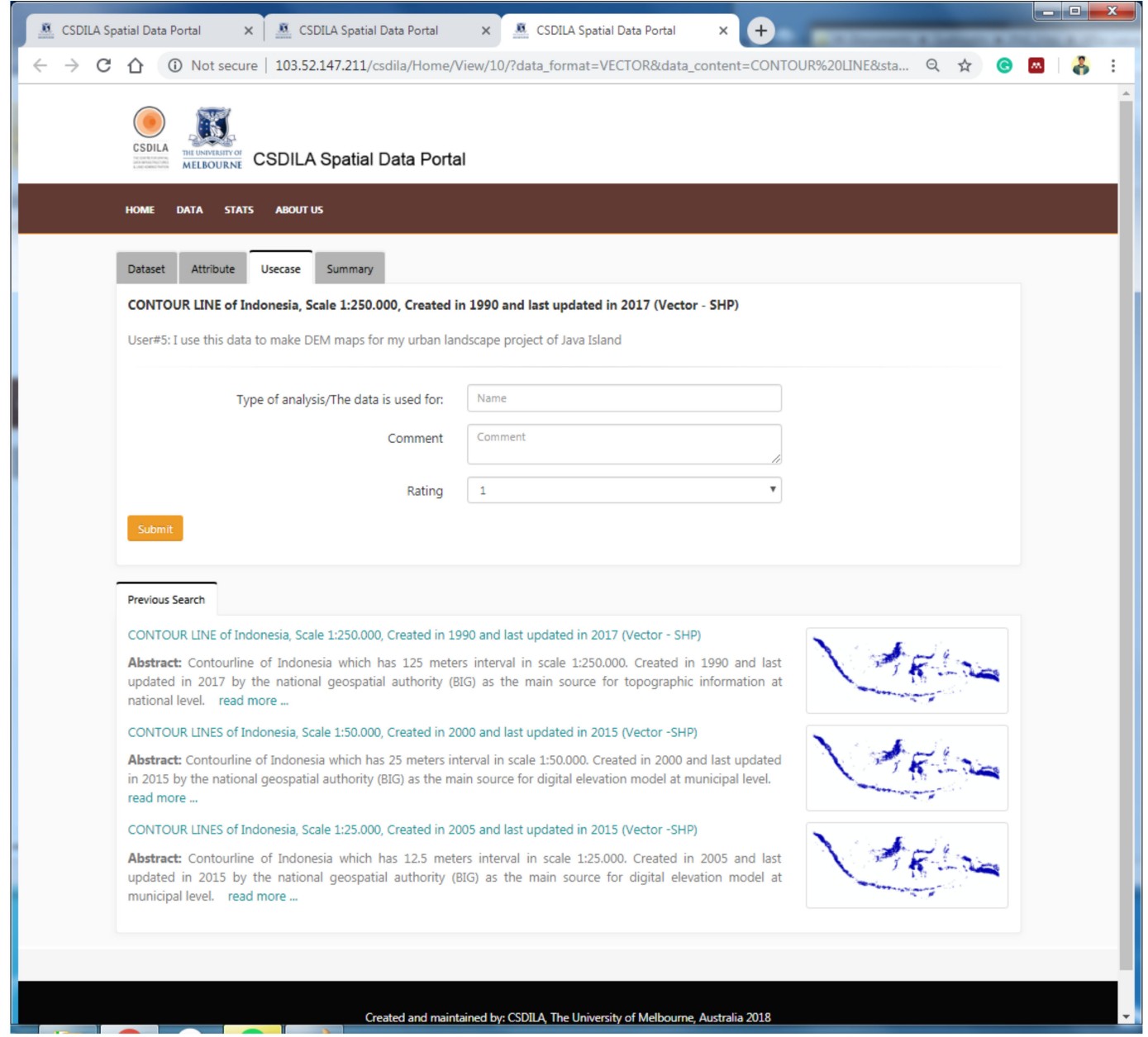

**Figure 8.** Metadata page (Usecase tab).

### 2.2.6. Summary Metadata Page

The last tab provided on the metadata page is the summary tab. This gives users a summary of the spatial data mainly about the production process and also includes the data's sources, production method or technology and the organisations conducting the process, as illustrated in Figure 9. The previous search window is also presented on this page.

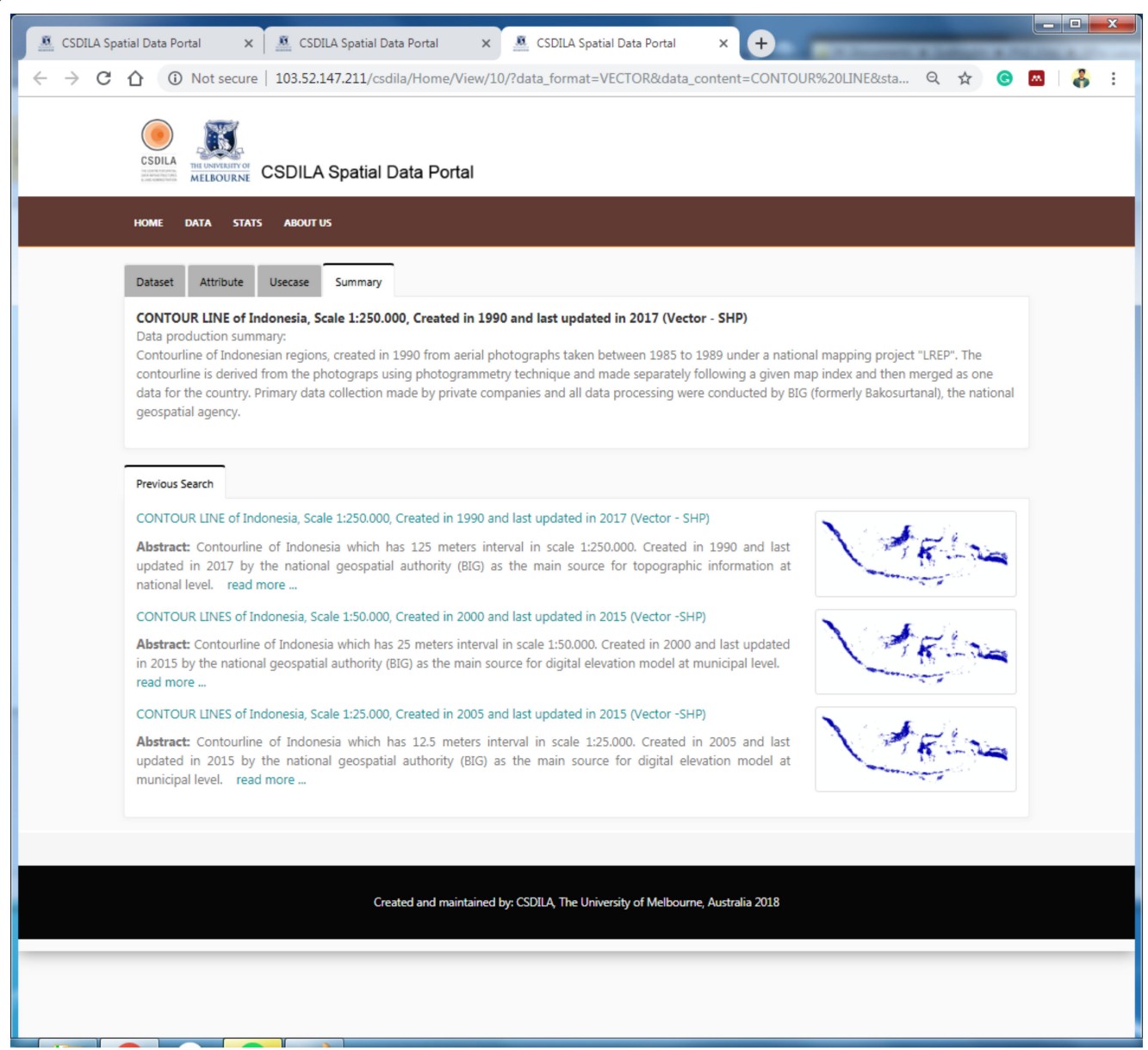

**Figure 9.** Metadata page (Summary tab).

### 2.2.7. Web Map Service Page

The web map service (map preview) page is designed to show a preview of the selected spatial data in an OGC web map service format. Users can activate the page by clicking the thumbnail (graphical information showing the geographic extent of the data) either on the results page or on the dataset metadata page. The page contains a map window and a legend window that allow users to play with and familiarise themselves with the data, as illustrated in Figure 10.

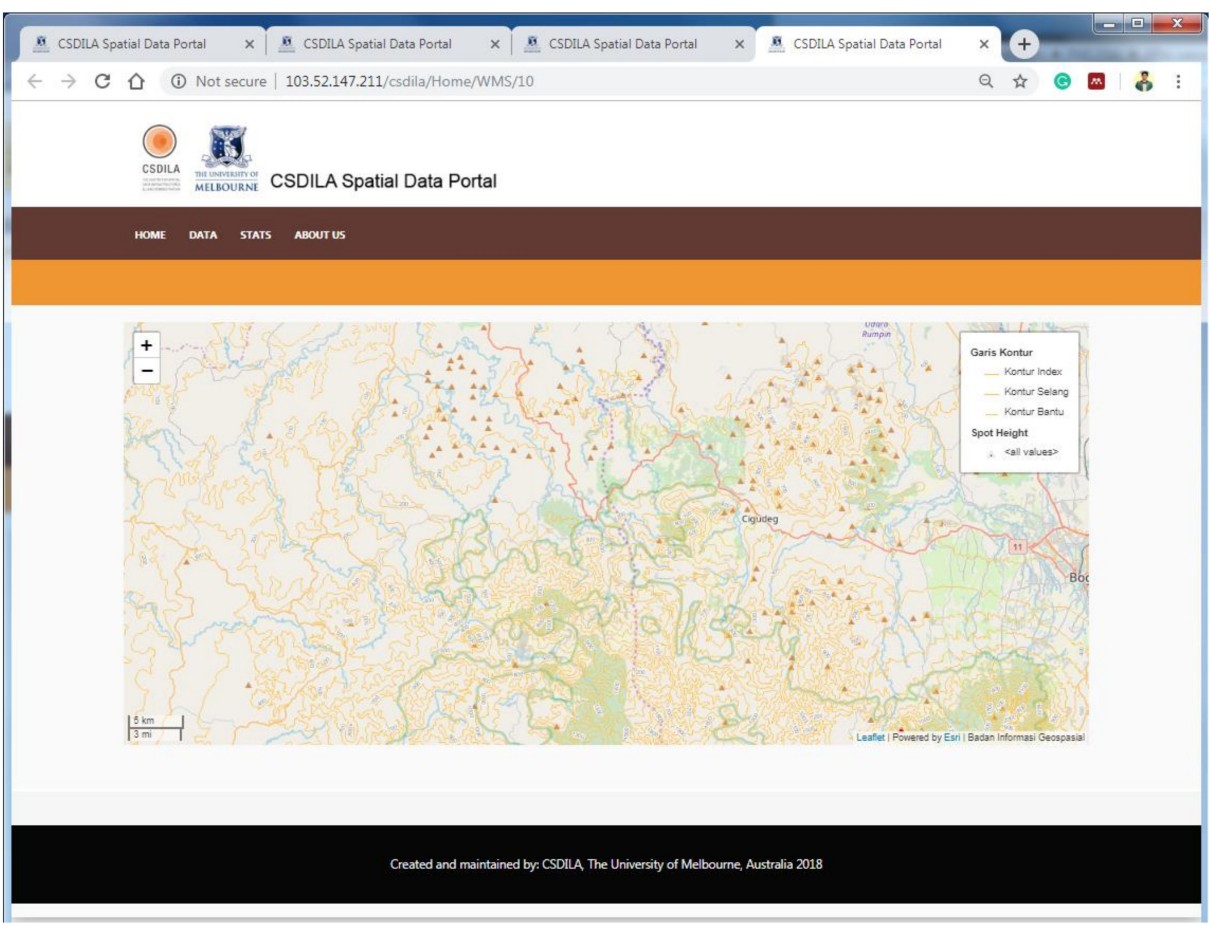

**Figure 10.** Web map service page (Map preview).

*2.3. Evaluating*

The evaluation follows a process similar to that used by [23] to close the loop in addressing the usability problems identified by that research. The purpose is to maintain the consistency of the methods and indicators between the two studies.

Like [23], the data-collection methods for this evaluation purpose include TAP usability testing combined with semi-structured interviews. Eight spatial data users, including four participants of [23] from various professional backgrounds, participated in the data collections: A planner, three spatial data professionals, and four professionals with computer science backgrounds. Previous research [25] suggests a small number of participants with a long period of participant–researcher interaction for this type of study. They all work closely with spatial data in different areas and various projects in different organisations. Quantitative studies that focus on statistics, using many participants and little, if any, participant–researcher interaction, are not recommended by [25].

Each participant is assigned the role of a project member who has a responsibility for preparing spatial data for a flood evacuation plan. They are instructed to find and select required spatial data to be used as the major data source for developing the evacuation plan. The scenario is accompanied by a set of instructions issued before the commencement of the TAP and the interviews in a written format.

The data were analysed using a combination of protocol analysis (verbal reports from research participants), and transcript analysis was implemented for data analysis. Here, any comments or expressions are considered to have the same weighting potential, even though any comment is only submitted once by a user. The level of the importance of a statement is not based solely on how often it is revealed but also on how much it is discussed during the interaction between the user, the prototype and the interviewer. The

results were interpreted to determine the usability of the proposed prototype based on the following criteria, which are similar to [23]:

1. *Effectiveness*. The extent to which spatial data users successfully discover and select spatial data.
2. *Efficiency*. The resource (time) spent by spatial data users to successfully discover and choose spatial data.
3. *User satisfaction.* Emotions, feelings and experiences of spatial data users during the data discovery and selection processes.

The evaluation was also designed to identify usability problems and user expectations, which will be helpful in future development following the iterative nature of the UCD approach.

## 3. Results and Interpretation

The data from the TAP and the interviews were analysed to extract the information required to evaluate the usability criteria of the prototype: Its effectiveness, efficiency and user satisfaction. It is also required for identifying problems and gathering user expectations.

By conducting protocol analysis for the TAP data and script analysis for the interview data, the results indicated that the usability of the prototype for both spatial data discovery and selection is higher than the usability of the established metadata systems. However, the interpretations of the results should be carefully discussed and explained before they can be concluded.

Detailed explanations about the results and the interpretations of the usability evaluation for the prototype are explained in the following sections.

### 3.1. Effectiveness

The effectiveness of the prototype is measured by the successfulness of the participants in achieving the objective of the following tasks: (1) To discover the potential data and (2) to select the data to use for the given scenario, as explained in the following sections.

### 3.1.1. Effectiveness in Spatial Data Discovery

The usability of the evaluation results in the identification stage suggested that the effectiveness of the spatial metadata is a combination of the usability of the user interface and the metadata itself. Therefore, it is essential to address the issues and problems existing in both. The results from the usability evaluation conducted in this study confirm [23] and allow us to have a deeper discussion.

The following phrases indicate that the participants could discover the required spatial data from using the prototype:

"It's good, the searching is easy and straightforward, with some options of searching. I can see the results are relevant."

"The titles in the result page (are) really helpful to quickly find the most relevant data."

"The titles in the result list. They give me more information than any other portal I've used before."

Most participants had a clear understanding of the discovery tools, especially the filter-based one.

"The information (used) in the filters in (the) homepage are important."

Their searches were straightforward. They used the required filters to expect that the system would respond by giving them the spatial data matching their submitted criteria.

"The search was done in no time (because) it can be easily filtered based on user interest and the results (titles) are quite clear and consistent."

They were impressed with the results, especially the titles, because they found that the titles contained information that they could use to find the most relevant data for the project quickly. This information included what the data are about, who created the data, when and where the data were created and how the data are stored. The information was helpful for them, enabling them to quickly recognise the potential of the data for the given project. It can be used as the basis for data selection if they have no other information or make an instant selection.

> "(The) titles are very helpful since they contain all the required information (what, where, when, who)."

Due to irrelevant results to the submitted search criteria and the absence of any filters, the scale of information presented to the participants, and thus, the inconsistency of the information presented in the titles, meant that hardly any of the participants could discover all the data they required within a given time. If users had more time and decided to continue the search, they might find the data they needed. However, this was quite a frustrating process due to the issues mentioned above with both the metadata records and the user interface.

The effectiveness of the prototype is as expected because they were designed and developed to address the issues. However, another fact can be discussed regarding the effectiveness of the prototype compared to the existing metadata and data portal regarding their production and development.

The metadata records presented in the two established portals in [23] were created and maintained by different organisations, and another organisation supports the portal. Those organisations might have both used the same metadata standards, ISO 19115:2003, when they designed their metadata as well as when they created the portal. However, from the evaluation in [23], it was found that metadata records vary from one to another, and the information does not necessarily support the discovery system or vice versa. Meanwhile, in this study, both the metadata records and the discovery system were designed and developed simultaneously and then synchronised. Therefore, the information contained in the metadata records supports the functionalities of the discovery system and vice versa. This fact indicates that the effectiveness of the spatial metadata for spatial data discovery can be significantly improved by addressing both metadata records and the user interface (discovery tools) issues simultaneously and by synchronising the information in the metadata records with the functionalities in the discovery system based on user requirements and expectations.

### 3.1.2. Effectiveness in Spatial Data Selection

Participants found that the prototype gives them the information needed to decide whether to use the data for a selected project. However, these results should be interpreted carefully to determine the effectiveness of the prototype. The following quotes might be helpful to begin with:

> "Information presented in the metadata page (is) important and relevant, but some key information for a specific type of data, e.g., interval for contour data, should be presented in the table (view)."

> "All and all, the information presented in metadata is clear and necessary."

> "Usecase gives me an idea what dataset might be suitable and what other people said about it."

The above quotes indicate that the information presented in the prototype was valuable and understandable for the participants when selecting spatial data. However, the quotes do not allow us to determine the effectiveness of the prototype for spatial data selection. In this case, effectiveness is defined as a participant making clear and confident decisions rather than vague ones. The following quotes contain participants' thoughts about their decision-making process that could be useful for detecting the effectiveness of the prototype for spatial data selection:

"I am confident with the information to make a decision to use the data or not."

"But I need the reference to the original data source (such as USGS or other institution) to make sure that the data is legit and comes from a valid source."

"If I only have the metadata information, I am not confident to make a decision and I definitely need to look (at) the data."

The above phrases about the participants' decisions can be divided into two groups: (1) Confident and clear; (2) conditional. The former group is clear because participants would decide based on the information in the metadata. The latter group indicates that participants might have decided but might also need other information potentially beyond the presented information. The following statements indicate this:

"This authoritative information (publisher and license) is important for the validity of the data."

"I know that the data is produced and published by a valid organisation, so there is no reason I cannot make a decision whether to use or not the data for my project."

The phrases "authoritative information," "validity" and "valid organisation" refer to the participants' specific knowledge of the organisations presented in the metadata. The phrases indicate that they can only make a clear decision if they know the validity of the data's producer. In other words, they are not able to decide if they do not know this information.

The following indicate that the information presented in the metadata is still insufficient for this participant to make a firm decision:

"But I also need to check the data once it's downloaded to make sure that the data would be fit to this project."

"If I only have the metadata information, I am not confident to make a decision and I definitely need to look (at) the data."

Instead, they insist that a direct experience with the actual data or information from other sources is necessary as part of their decision-making process. Similar issues, among others, were also found in [23] for the established spatial metadata systems.

The participant stated the phrases mentioned above before they found other prototype capabilities specifically designed to address the issues. Additional information was presented to the participants, such as the usecase tab, data ratings or new capabilities, such as those appearing on online shops (e.g., a list of relevant data, previous searches and web map services for a data preview). When they found and explored the web map service as a map preview, the following was mentioned by the participants:

"The information in this page (web map) is somehow more important to me because no matter the quality of the metadata, I (would) need to check the data."

"This (webmap service) is a must, because I don't have to download and check the data to play (with) and look at the data. It saves time because I don't have to download a lot of data and contact a lot of persons to know about the content of the data."

"It (web map service) gives me a complete sense of the data. For example, if I need a parcel data, and if I can have a look into that specific area without buying or downloading it first, I would be very happy, and I think it (i.e., the web map service) should be a metadata element."

The above phrases indicate that the participants still need to see and check the data to get the information that cannot be provided textually in metadata. To some extent, the spatial data web map service can fill this need by providing a live preview of the spatial data without the need for downloading or buying the data. To some extent, the map

preview is more important than other information presented in the metadata regardless of its quality.

However, not all participants agreed with this. One participant had a different opinion about their decision-making process, saying that they still might have to look at other additional information regarding the data from other sources. This can be seen from the following:

> "I am okay with the information in the metadata and the presentation, but I might have to look on other information from other source(s) as additional information."

This participant mentioned the specific requirements regarding the spatial data attributes such as the attribute type, statistics, and other quality standards, e.g., quality evaluation methods and the results.

The following phrases were obtained from participants' verbal comments when using the usecase tab and saw the rating.

> "Usecase and rating help me to have peace of mind about the fitness of the data for my purpose."

> "The usecase is important. If I, let's say, find out that the data had been used by figures or organisations that are reliable, then it will give me the confidence to use the data. It enhances the quality in a sense of reliability."

> "It's complementary, but it (i.e., usecase) is important because it gives me objective reviews based on user actual experience, (whilst) the information in the metadata page presents the facts about spatial data. I will consider both (categories of) information to make a decision."

> "I like the rating and number of downloads (because it may suggest that) the higher the rating and number of downloads might be, (the) more relevant or reliable (it might be)."

These quotes indicate how the usecase tab and the user reviews on spatial data could be essential to the participants' decision-making process. The confidence level of the participants improved when they had this information. However, participants also mentioned the consequence of the rating and the user reviews if they were not presented properly. The planner said the following:

> "The persons who give a rating and download the data can be from different backgrounds with different interests. It would be better and make more sense if the rating and download are presented in a graphics showing groups of users and numbers of download and uses for each group."

According to this planner and the others, the user review, the rating, and the number of downloads should be presented in detail and should include information about the type of project and the group of users. Otherwise, the information can be useless at best and misleading at worst because they might get the wrong impression about the suitability of the spatial data for their projects.

Other additional information providing insights about the prototype is the spatial data relating to the participants' searches. It is one of the implementations of online shopping features, as requested by users in the user requirements:

> "That would give users useful information about the possibility and availability of other data that might be required for their projects. It will save time and provide insights to users in (the) data searching process."

The participants confirmed this idea, and their explanations are fascinating. The related-data feature potentially saves time for them and provides other data that might be useful for their project. It might also give the participant insight into the data content by looking at other relevant data in the list. Again, this information should be carefully prepared because it can be misleading if the additional data are not pertinent or are

wrongly determined. Having the results and interpretation of the TAP and the interview data featured above, we can identify several findings that should open a discussion relating to the effectiveness of the spatial metadata for spatial data selection.

Most of the negative comments in the first stage were due to the data's inaccessibility or the lack of a sample or preview and the absence of additional information in the data as evidence of its reliability. By providing access to the web map service as a data preview and to user-related information (e.g., reviews, a data rating), more positive comments were found on the data-selection activities conducted in this usability evaluation. This indicates that the effectiveness of the prototype for spatial data selection is higher when compared to the established metadata systems.

However, apart from the quality of the metadata records and the user interface, the effectiveness of the spatial metadata for data selection is subject to individual knowledge (i.e., spatial concepts and the reputation of data producers or distributors), experiences with the spatial data (so they can use this when they check the data), the availability of spatial data and also a participant's trust of the metadata, as reflected in the following:

"What if it's rare data that only a few people know about?"

"What if it's the only available data for that specific theme?"

"I think I might need additional information from other sources."

Nevertheless, the validation stage provides the required data and information for evaluating the effectiveness of the user-oriented metadata prototype for spatial data selection. They indicate that the effectiveness, when compared to [23], is improved. Additional information about the capabilities of the prototype—such as user-related information (the usecase tab, a data rating, the number of downloads), related data to the current search and the web map service as the data preview—proved helpful in increasing the participants' confidence in deciding whether to select the data for their given project or not.

### 3.2. Efficiency

The results from the usability evaluation indicate that the efficiency of the prototype for spatial data discovery and selection is higher than the efficiency of the metadata systems in [23]. The following sections describe the results and interpretations of the TAP data and the interview data from the usability evaluation of the prototype.

### 3.2.1. Efficiency in Spatial Data Selection

The efficiency of the prototype for spatial data discovery is determined by observing the time spent by participants on the discovery task. The participants' comments are then analysed during the data discovery process.

The researchers' observations during the evaluation show that all participants were successful in immediately discovering the required spatial data for the given scenario. This was expected because the metadata records and the user interface were specifically prepared following the user's needs and expectations, as presented in the user-oriented metadata specification [23]. The researchers prepared the content of each required element used by the participants as the search criteria as well as the functionalities of the user interface to support the discovery process. As explained in the prototype development section, both the metadata and the user interface were designed, developed, and synchronised simultaneously.

The following indicates the highly efficient discovery process conducted by the participants:

"I can easily filter the search and the results (i.e., titles) are quite clear and consistent."

"It's good, the searching is easy and straightforward, with some options of searching (keywords, filters). I can see the results are relevant."

As mentioned by the participants, they found it easy to use the discovery tools. The criteria are straightforward, and they can submit all the criteria required for their search to find the given spatial data. The results were even better, especially the titles, when they contained the basic requirements for spatial data discovery and when they were presented consistently. This can be identified from the following:

> "Titles contain all the required information (what, where, when, who) and the consistency is important."

> "The titles saved my time to get basic information about the data without the need to open the metadata page."

According to the participants, the titles contain information answering all questions regarding the who, what, why, when, and how questions related to spatial data. All the following are presented in titles with a reasonable length of words: What the data are about, where it is covered by the data, when the data were produced or last updated, who the producer is, and how the data are conditioned (represented by the scale and data format).

The titles alone are sufficient for the participants to make a quick selection of spatial data. The abstract provides them with additional helpful information for more detailed descriptions of the data. All the information is presented consistently, making it easy to follow and read, even if the results consist of a lot of data.

However, for similar reasons, the participants agreed that the abstract should not be presented because the titles already provide them with the required information. They instead prefer concise yet valuable information, such as keywords, to be presented alongside the titles to glimpse the context of the spatial data. They also like the abstracts to be on the metadata page to contain more results.

Regarding these evaluation results, the prototype's efficiency for spatial data discovery is more significant than when compared to the established metadata systems. The reason is apparent: tThe spatial metadata records were prepared to provide all the information (element) required by spatial data users to discover spatial data. The user interface offers the functionalities necessary, such as filters or search criteria, to support the discovery. Another reason is that the results (title and abstract) were consistently prepared following the user requirements in the specification to make it easier for the participants to recognise critical information when identifying potential data for the given project. However, the established metadata systems did not provide similar information and did not have the consistency seen in the spatial data discovery prototype titles.

### 3.2.2. Efficiency in Spatial Data Selection

The participants found that the required information for selecting spatial data was easily identifiable and understandable. They were instantly able to find critical information and to make their decision regarding the spatial data. The participants found it easy to decide whether they wanted to select and use the data for the given project based on the presented information, which is reflected in the following:

> "I'm quite impressed with the website, as its clear and informative, straightforward, and the information is very useful and enough for me to make a decision whether to use or not the data."

> "I think the system is easy to use. The information is understandable."

However, there were some negative comments from the participants, mainly about the user interface. This can be identified in the following:

> "I think the information is useful to know about the data, but I didn't know that the titles (in the result list) can be clicked to find metadata. I think you should give a clear instruction on the result page."

> "I didn't notice about that (the web map service) at the first place, so it requires a sign or instructions to make users notice it in the first place."

There were similar phrases in the data that allowed us to identify the prototype's efficiency for spatial data selection.

As can be seen from the above, the efficiency of the metadata was considerably high because the participants mostly gave positive statements regarding the importance and relevance of the presented information. There were some negative comments regarding the information (or metadata contents). Still, such comments are more likely to be used for improvements, which is explained later in the user satisfaction and expectations sections. However, there were comments from the planner worth noting that questioned the relevance of some information to their search:

> "I don't know this (contour type: index contour, additional contour). For me, contour line is just one, contour."

> "I don't know this dataset type."

It does not mean that the information is not there, but the additional contour types were confusing for him because he would normally only look for the main contour data. As for the dataset type, when it was explained by the researcher during the interview, they agreed that the information was useful. However, they requested a different description be displayed.

In terms of the user interface, the design and functionalities of the prototype in presenting the metadata make it easier for the participants to quickly identify the information. Credit can be given to the table format for succinctly presenting information, i.e., using a limited number of words while still being informative.

A special credit can also be given to the web map preview. Because most participants required seeing and checking the data before making a decision, most stated that the preview would save them time when deciding whether certain data could be used for the project:

> "I think it is good to have it (the web map service) because it will save my time to download and check the data (also the money if it's not free) to check whether the data is valuable for my work or not."

> "Web map service is a must, because I don't have to download and check the data and can play and have a look at the data and save time since I don't have to download a lot of data and contact a lot of persons to know about the content of the data."

The data, along with the user review, allows participants to identify the web map service as making the data-selection process more efficient (regarding both time and money). Specifically, they could play and check the data first without having to download and buy it. They could also save time by not having to get the information from other sources.

Participants found that the previous search window was helpful because it provided them with their current search at the bottom of the main metadata page. They could jump to other results from the window without needing to perform another search. This saved time in the selection process.

From the above descriptions of the evaluation results, this prototype's efficiency for spatial data selection has improved compared to [23], where the participants had difficulties with identifying and understanding the information presented in the metadata. It is not that the participants found no problems with the prototype, but the prototype allowed them to make a quicker decision and, to some extent, a cheaper one.

The results show that the efficiency of the spatial metadata for spatial data selection is determined by the following: (a) The completeness of the metadata against the users' selection criteria, (b) the relevance of the content (information) to users, (c) the presentation style and (d) the functionalities, which should make it easier for users to recognise the information.

### 3.3. User Satisfaction and Expectations

User satisfaction is related to the participants' emotions, feelings and experiences when they worked and utilised the prototype during the usability evaluation process. It was not easy to measure and determine them. However, we can use the phrases to detect the participants' satisfaction and their expectations of the prototype, which is exemplified below.

"This is a good way of (presenting metadata), based on my previous experience with metadata with other software like ArcGIS, Mapinfo and other types of (systems). This approach gives (a) more comprehensive way of presenting metadata."

Similar notions can be found in the data, which indicate that the participants were satisfied with the information presented in the prototype and the user interface (i.e., presentation and functionalities). A participant who happened to be an ex-national open data manager mentioned that they were impressed with the prototype because it was informative, clear and easy to use. He also suggested that the prototype should be commercialised or implemented as a product.

The participants also gave many comments about the prototype that can be identified as their expectations to improve the prototype. Table 1 summarises the user's expectations of the prototype to better help them in the spatial data discovery and selection process.

**Table 1.** User feedback on the prototype.

| User Interface | Metadata Records |
|---|---|
| ✔ The default searching tool is the simple one (i.e., keywords) and can be extended to the advanced filter and spatial-based search tool | ✔ Additional information, i.e., the validity of the data (valid, used, or obsolete) |
| ✔ A detailed data format for both vector and raster data, e.g., shp., cad., and cov. for vector or tiff., jpeg., png. for raster | ✔ Detailed information about previous users and the number of downloads, i.e., type, project or field of work |
| ✔ Clear instructions for the spatial-based search tool | ✔ Detailed information for data rating based on pre-determined criteria as on e-commerce web pages |
| ✔ Clear instructions for opening the map preview (web map service) | ✔ Additional information on the attribute data, i.e., statistics, data attribute type |
| ✔ Clear instructions for opening the metadata page from the search results page | |
| ✔ Alternative presentation of the titles | |
| ✔ Detailed data rating based on pre-defined criteria per spatial data quality or user-specified criteria | |
| ✔ More functionalities in map preview, e.g., object identification and statistics for the attribute information | |

From Table 1, we can identify that the user's feedback is mainly related to the user interface. The participants considered the information sufficient and deemed that fine-tuning should only be made to the functionalities and the presentation. As for the metadata records, most of the participants agreed that the detailed user reviews, the number of downloads and the user data ratings are necessary. It was mentioned by a participant in an earlier section that the user-related information was important in assisting their decision but only when this detailed information related to the user and the project given.

Compared to the user satisfaction of the established metadata systems in [23], the participants were more satisfied with the prototype. Two participants who performed the evaluation for both metadata searches commented on the prototype more positively compared to the previous ones.

The result is as expected because the prototype was designed and developed based on the results of [23] and was catered to meet the user's requirements and expectations of spatial metadata and the user interface. However, users will never be fully satisfied, and that is normal. They will always find new needs and new expectations. That is why usability is something that should be evaluated regularly to ensure that the metadata, as well as other products, meet at least most of its user's expectations.

### 3.4. Usability Problems

Like the user's expectations, usability problems were identified because participants will always have new needs and expectations of the metadata. However, unlike the issues identified in [23], the participant's findings during the prototype evaluation did not prevent them from discovering and selecting the spatial data.

During the evaluation process, the most noted problem was the unclear instruction on how to click or activate the essential features for the spatial data selection, such as the map preview (web map service). Most of the participants did not know about the preview because it was not presented with the metadata. Participants who found the preview either gave more attention to the mouseover style link or accidentally clicked the link located in the thumbnail picture (i.e., geographic coverage). A similar problem was also found regarding the spatial-based search tool, as illustrated in Figure 11 This problem is considered high risk because the map preview has proven to be a crucial tool, and this issue could jeopardise the participants' decision-making process.

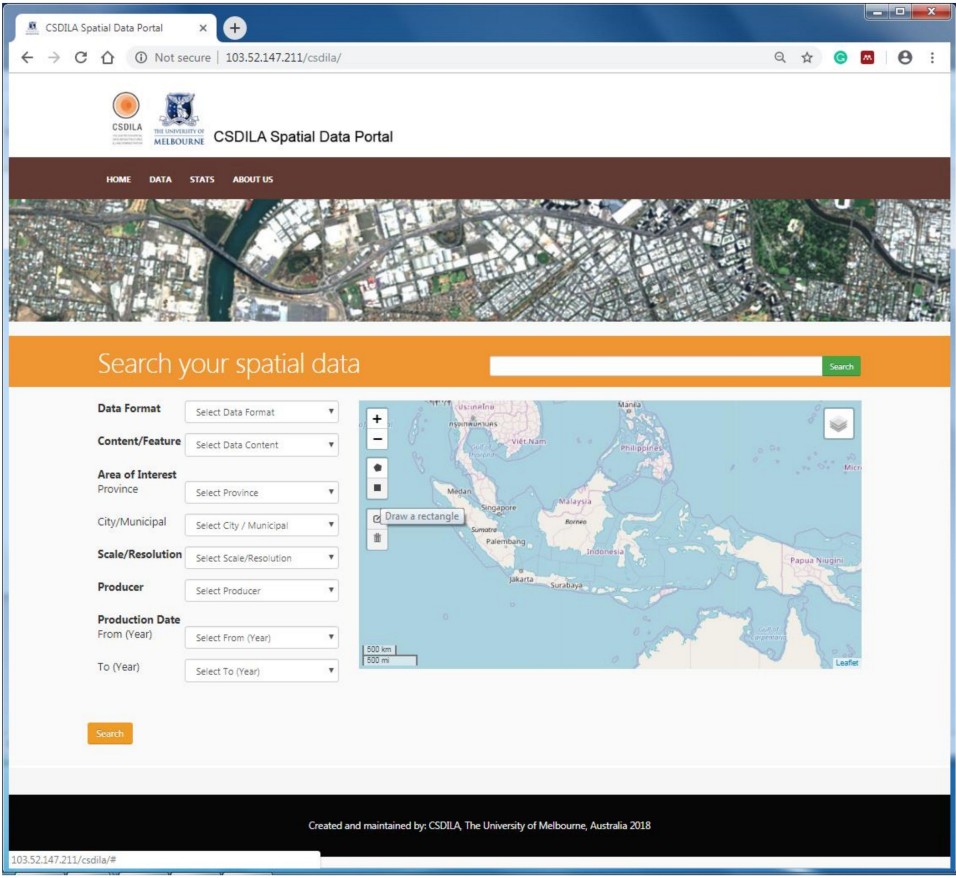

**Figure 11.** Unclear instruction for the spatial-based search tool.

A similar instruction problem was also found on the search results page, where participants did not know that the titles could be clicked to open the metadata page. That is, again, considered a high-risk problem because users with no experience with the website or with spatial based data searching tools would stop there and fail to continue the process. This is a typical usability problem with a user interface or a website [10].

In terms of the information from the metadata records, participants found some of the information irrelevant. This happened to the planner in an earlier section when he looked at the content of the contour data. It did not necessarily prevent them from getting the relevant information, (i.e., the main contour line), but it took some time to identify the critical information.

Other problems identified during the evaluation process were mainly related to the participants' expectations, such as regarding the user data rating and the user reviews, where the participants requested detailed information rather than just a general statement, such as good or bad, as mentioned previously. All the identified problems were significant, and they should be addressed in the next development stage of the spatial metadata and the user interface.

Having explored and interpreted the usability evaluation results of the newly proposed metadata prototype and user-interface prototype, the following section discusses the results.

## 4. Discussion

Using the three attributes of usability—effectiveness, efficiency and user satisfaction— the results indicate significant improvements in usability across all criteria. The results showed that participants were interested in the spatial data search tool, especially the criteria (filter)-based option. This allowed them to narrow down their search as early as possible. Participants also found that the results page, which listed data matching the search criteria, was very pleasing and easy to understand. The consistency, length and content of information presented in the title and the abstracts were helpful for users to quickly recognise what the data were about. Accordingly, they could promptly preview the potential data for the given application before clicking them to check the detailed information. The list of relevant data to the search was also interesting to participants. They could use the information to get to know other data beyond their knowledge that might be useful for their projects. In terms of efficiency, this feature saved time while searching for other required data. This finding shows the potential of linked metadata with subsequent metadata development.

Moving on to the main metadata page, the results found users agreed that most of the information presented in the dataset tab was helpful for them to assess the quality of the data. The tabular presentation of the metadata made it easier for them to instantly identify the information most relevant to their interests, e.g., data content with a high level of detail, positional and attributional accuracy and contact details. Information about user reviews, data ratings and the number of downloads was also exciting and important to participants. They can use these as alternative measures of data's quality and fitness for use. Such information was unavailable in the usability testing done by [23]. However, participants requested that the information be presented in detail for them to obtain the relevant information. Accordingly, the form provided on the usecase tab should have the exact requirements so users might provide their experience with the data correctly.

Another noteworthy finding from the results was that participants were interested in the web map service feature of the data. This gave them the chance to access the data, thereby giving them the confidence to assess its suitability. They could check the data, something missing in the usability testing of the established metadata systems done by [23].

The users' reactions and comments indicate that they were satisfied with the prototype, apart from the several requests for other improvements, which should be expected from a usability study.

Lastly, the results validate the proposed synchronised development approach for the spatial metadata and the user interface in increasing and maintaining the usability of the metadata for data discovery and selection. Because the usability of the metadata is a combination of the metadata and the user interface, a synchronised development approach, with users at its centre, is required to improve and maintain the usability of the metadata.

## 5. Conclusions

This paper evaluates the proposed improvements to metadata systems in the SDI literature through prototyping and engaging with end-users. A metadata system prototype for selected spatial data from the Indonesian Geospatial Agency was accordingly created, e.g., road networks, contour lines and administrative boundaries, along with a web portal for the spatial data discovery tool as a user interface.

The prototype was then evaluated by conducting TAP usability testing and semi-structured interviews with eight spatial data users with varying backgrounds. They were asked to complete a series of tasks in a given scenario within a given timeframe and to think aloud, giving verbal expressions during the TAP.

The result of this study indicates that maintaining the consistency and relevance of the information presented in metadata and consequently improving the usability of spatial metadata can be achieved by proposing more stringent obligations to the elements required by users. This includes intended use, data samples and replacing the free text domain of crucial elements, such as title and abstract, with a compound domain comprising content from other selected elements following the user requirements. In addition, we added new metadata elements, i.e., a user data rating and the number of downloads.

The results also validate the proposed synchronised development approach for the spatial metadata and the user interface in increasing and maintaining the usability of the metadata for data discovery and selection. In the distributed geospatial data catalogue platform, the spatial metadata records and the spatial metadata catalogue are not necessarily created and maintained by the same person or organisation. Neither is the catalogue gateway, where the user interface and its functionalities are defined and developed. Because the usability of the metadata is a combination of the metadata and the user interface, a synchronised development approach, with the users in the centre, is required to maintain the usability of the metadata.

Together with [23,24], the current paper collectively contributes to SDI knowledge in several ways. First, UCD is a valuable method to be implemented in a research context. Second, these studies collectively suggest that the spatial metadata standard, ISO 19115-1:2014, should be improved by extending it with additional metadata elements, changing the more stringent obligation type, and replacing the free text domain with the contents of other relevant elements, as required by spatial data users. The user-oriented spatial metadata discovery profile, as an extension of the ISO19115, can increase the usability of spatial metadata.

**Author Contributions:** Conceptualisation, Mohsen Kalantari and Syahrudin Syharudin; methodology, Syahrudin Syahrudin; validation, Syahrudin Syahrudin, Mohsen Kalantari and Abbas Rajabifard; formal analysis, Syahrudi Syahrudin; investigation, Syahrudin Syahrudin; resources, Mohsen Kalantari; data curation, Syahrudin Syahrudin; writing—original draft preparation, Syahrudin Syahrudin; writing—review and editing, Mohsen Kalantari; visualization, Mohsen Kalantari; supervision, Mohsen Kalantari, Abbas Rajabifard; project administration, Hannah Hubbard; funding acquisition, Mohsen Kalantari, Abbas Rajabifard. All authors have read and agreed to the published version of the manuscript.

**Funding:** This research was funded by Australian Awards Scholarships and Australian Research Council grant number DP170100153.

**Institutional Review Board Statement:** The study was conducted according to the guidelines of the Declaration of Helsinki, and approved by the Human Ethics Advisory Group of the University of Melbourne (application number 1545774.3 and 03/10/2018).

**Informed Consent Statement:** Informed consent was obtained from all subjects involved in the study.

**Acknowledgments:** The authors would also like to thank all the members of the Centre for Spatial Data Infrastructures and Land Administration (CSDILA) and the Centre for Disaster Management of the University of Melbourne for all the discussions and enjoyments. We also acknowledge Hardy Subgyo for his input towards the prototype development.

**Conflicts of Interest:** The funders had no role in the design of the study; in the collection, analyses, or interpretation of the data; in the writing of the manuscript; or in the decision to publish the results.

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
