# Peer review of "Synchronising Spatial Metadata Records and Interfaces to Improve the Usability of Metadata Systems"

_ijgi, doi:10.3390/ijgi10060393_

Round 1

Reviewer 1 Report

I have no more comments on the substantive content of the work. However, I have comments on the editing of the article:

  • a lot of unnecessary blank lines, e.g. 79, 80, 137, 141, 148 and so on;
  • something strange happened between lines 581-587, especially line 583.

I believe some of the issues will be fixed by the MDPI editors. But special attention of the Authors is needed in chapter 4.

Author Response

Thank you for your feedback. We removed the unnecessary spaces/lines in the paper. 

Reviewer 2 Report

From the authors' notes, and being the layout changed (see by ex numbers of lines), it is difficult to trace if and how previous suggestions have been concretely met in the new text. Then a new reading has been necessary.

I continue to think that the paper offers a very marginal contribution to the field and I try to demonstrate my opinion here following.

From this paper Introduction I read:

“Based on the user-centred design (UCD) methodology [25], [23] engages end-users to understand their view on the efficiency and effectiveness of existing metadata systems. [23] discovers significant gaps and usability issues in terms of spatial data discovery and selection. Building on [23], [24] engages users of various existing data portals to delve deeper into the usability issues. [24] offers improved design principles for the interface in metadata systems based on the engagement result. [24] also provides an extension to the ISO 19115 standard to assist the user needs for spatial data discovery and selection.  This paper aims to gauge users’ acceptance of the newly proposed spatial metadata profile and user interface by [24]. This paper first analyses the interface requirements concerning the extension. It then implements them in a prototype system. Finally, it evaluates the proposal by engaging end-users.”

First of all, [25] is a general book dealing with a lot of topics in doing research (Theoretical perspectives and research methodologies. Doing Research in the Real World, 2014): nothing else (more specific, more recent) to be cited with respect to UCD and UCD plus metadata? In any case authors should specify why they limit / focus their source citation to this title.

Letting aside this first observation, I have the possibility to read and verify the content of [23], i.e. Kalantari, M., Syahrudin, S., Rajabifard, A., Subagyo, H., & Hubbard, H. (2020). Spatial Metadata Usability Evaluation.

But as far as [24], it seems to be a thesis and being under embargo, so that I could read only the abstract of this work. My feeling is that the present paper presents (a part of) the work done in [24]. It is then very difficult to evaluate the real original contribution of this present work with respect to [24].

This continue to be, in my opinion, a major issue in this paper.

Moreover, I observe that:

  • the references are too limited with respect to an enormous literature: I invite authors to enrich them
  • not having a statistical sample of users does not mean that the results must be only qualitative, graphs, percentages,  synthesis are possible and welcome
  • it is not clear if and how this limited case study can be generalized

Author Response

Thank you for your comments. Please see below for our response: 

  1. As suggested, we added more specific reference for the UCD method [26], which complements [25]. This paper focuses on usability in information systems and relevant to the topic of this paper.
  2. We removed [24] and replaced it with a paper in-press from it. The number for this new paper remains [24], though.
  3. To the best of our knowledge, we have included relevant literature in this paper by building on their contributions to knowledge and gaps. 
  4. As we have clearly described, this is a qualitative study involving eight participants, 4 shared with [23]. The participants have shared their experience with an improved metadata system using TAP, a guided interview. Graphs or percentages do not lend themselves to representing user experience.
  5. This study has been undertaken based on the UCD. We have first engaged the users to give us feedback on the mainstream metadata system [23]. We then verified the feedback by engaging another set of users [24]. We have implemented the feedback from [24] and evaluated it in this paper through the first set of users [23] and additional users. This is a robust methodology, and the results are reliable and can be generalised to metadata systems.

Reviewer 3 Report

The paper is now clear and can be accepted.

Author Response

Thank you for your review.

Reviewer 4 Report

Thank you for the renewal of this contribution and becoming more pragmatic with "Synchronising spatial metadata records and interfaces to improve the usability of metadata systems". 
The testing of the prototype with 61 samples seems to be enough to receive a trend for the acceptance and applicability of the prototype. Furthermore it highlights the importance of the user interface design and functionality design. 

Still, the paper does not mention the main principle of SOA, which is used by the computer-2-computer communication. Because of the focus on the application tier and the presentation of metadata in the application tier the embedding of SOA seems not to be appropriate anymore. This contribution has its focus on the functionality of the user interface. 

Some spelling erros were observed. The contribution should undergo an appropriate proof reading. E.g. in figure 2 some spelling errors exist, or in line 583 the last sentence is part of the conclusions heading. 

In line 132 the attribute metadata page is shown. This is a good step forward, but for the semantic understanding links to the definition of attrubutes, vocabularies and/or ontology is missing.  

Thank you for your effort and valuable contribution.

Author Response

Thank you for your comments. Please see below for our response

  1. As you have identified, this paper is focused on the application tier, and SOA is not quite relevant.
  2. The paper is now professional proofread.
  3. We agree that semantic understanding such as links to the definition of attributes, vocabularies and/or ontology is useful. Since the study participants did not identify them, we did not include them in the paper.

This manuscript is a resubmission of an earlier submission. The following is a list of the peer review reports and author responses from that submission.

Round 1

Reviewer 1 Report

Dear colleagues, thank you for your contribution "a user-oriented spatial metadata system". The title and abstract correctly follows an important direction: usability of spatial geoportals or spatial catalogues, which are called "spatial metadata system" in your paper. 

The importance of metadata and spatial catalogues goes far beyond organising, sharing, discovering and using spatial data of a SDI, it can be the core component of spatial knowledge structuring in a near future. 

Your observation and discussion is absolutely correct: 

Free text in metadata leads to inconsistent and in many times irrelevant metadata. Although ML search mechanisms can increasingly make use of it, the contextual value of the content is low. Often there are few ressources to fill the free text metadata comprehensively. Controlled vocabularies and taxonomies help for many issues, but require commonly defined and available registries, where the definitions are available. 

The most productive search engines make use of these taxonomies already, please evaluate the search mechanisms of Bing, Google and Yahoo and their knowledge structure at schema.org.

In section 2.2 you descibe your methodology and the experiment design. Your list of "user-defined functionalities criteria" are proposed by ISO 9241. But I cannot read anything about a cross-check: haven`t you evaluated this list against existing search portals? This is a first main drawback. A short evaluation of common search engines will allow to relate to spatial requirements. 

Your experiment has been done with a set of users from different disciplines. Eight participants are quite low for a design study. Some of them seem to be biased by their profession, which lowers the trend analysis. You should definitely enhance your study and make your observation more reliable.

In the beginning you state that you are following the UCD approach. This procedure has a clear structure. i cannot observe this clear structure for your validation process. What is the starting point, what is the aim, where is the recursion?

At leat I have to hint you to a massive methodological mistake in your work: the main statement of your contribution is that user requirements need to be embedded in the metadata of the data, in order to create "user-oriented spatial metadata profile". This statement is completely WRONG according to the principles of Service-Oriented Architectiures (SOA; a lot of literature is available and missing in this paper). 

Allow me to explain: 

The SOA principles split an information system (according to OASIS) into different tiers. THis splitting allows for different modules and the creation of common valid and generic interfaces inbetween. This creates the needed flexibility to maintain the information system. Searching for content is part of an information system. 
Your proposal for an user-oriented spatial metadata profile creates a direct relation between the user and application tier into the data tier. This system cannot be maintained usefully. It is a methodological error. Please evaluate some references for SOA. 

Your proposal could work, if you focus on the application and user tier: metadata shall be created for applications and usability/user needs. This also supports your observations in table 1, which are metadata of these two tiers. 

In the end allow me to recommend some studies for the Google search system. With "https://datasetsearch.research.google.com/" Google has created a search engine for metadata. All requirements, needed vocabularies and used standards are documented. You should definitely consider Googles input for a spatial search engine. 

I recommend to restructure the paper, complement it with the missing aspects and follow SOA principles. 

Reviewer 2 Report

Paper addresses an important issue on how to meet user demands with Spatial metadata and raises one important aspect how to improve it: by addressing usability of the metadata portals.

The paper indicates that usability of the metadata portals is a major issue and this is not currently addressed in many portals. However there is no attempt to review existing spatial data portals which could give useful insights what developed prototype should contain.  A statement why this was not considered should be included. Either stating this should have been done or explanation why it was not considered useful.

Developed three different pages seems to interesting and provide useful information to the users (as indicated by the comments). If the service design methodology would have been used an iterative process would have probably provided an even better improved user interface. 

Service design is currently the most used methodology in designing services. It does not seem to be applied here. This is a major weakness in the methodology part and has a potentially important consequences how relevant the results are. It remains unclear what scenarios were used. These should be included. The test group and their relations to the scenarios used. It seems users were asked to perform certain task which not necessary would be what they normally would do. It is necessary to include description of the scenarios and explain what task the users were asked to perform. If the service design principle would have been used there should have been users already involved with the design of the service not only testing the outcome.

Figure 4 presents the system architecture of the prototype. In line 110 it is indicated that the figure would also present the requirements but these are absent in the figure. At current state the figure is too simple and does not give a realistic view how metadata portals function in reality. 

Table 1 is claimed to be users expectations but is is clear that this is not the case. Table 1 shows users' comments on the prototype. Table content should be changed to indicate whether the comment is suggesting improvement or not (currently the ok sign indicates everything was ok which was not the case)

Figure 13 presents the proposed new methodology by the authors. However the methodology is not explained and the figure remains unclear. Suggest to remove.

Reviewer 3 Report

This work aims to propose "a metadata system that is developed based on the requirements elucidated by engaging end-users".

In reality it seems to build prototypes (maybe of the metadata database and the user-interface) that are then tested to verify and collect the users' requirements.

The paper has been built on top of a cited work and the originality gap with respect to this previous work is not clearly declared.

The evaluation results are barely qualitative: quantitative comparisons have to be provided.

Lines 35-37: authors should better detail which are the elements of their research that are based on the work in [33] and, more important, which is the originality gap of the present work with respect to [33]. Moreover, in the Introduction they should clearly specify the issue(s) they wish to face. 

Line 50: the acronym UCD should be written in its extension this first time; please provide a due citation of the method source.

Lines 62-63: you mention here a prototype evaluation; which one of the two mentioned at lines 55-56? In general, having two prototypes create confusion also in the following: the reader is puzzled by identifying which one of the two prototypes are concerned in the various parts of the paper. By ex. at line 70 and the following … I’ve the idea that the two prototypes mentioned here correspond to the metadata database and the web pages to access and managed it; is this true?

Line 64: please provide a due citation of the TAP

Figure 3: it is not clear at all if and how the functionalities of the user interface are derived by users’ requirements

Lines 92-108: authors list here a set of requirements for the user interface but do not explain how they assure their realization in the own prototypal development

Lines 110-111: the figure 4 does not “illustrates … requirements for the prototypes”

Line 123: the prototypes (plural) were implemented as a website (singular): not clear

Lines 189-190: which is the evaluation performed in the first stage? The same at line 192: authors mention a “previous evaluation” which the reader cannot identify

Line 198: which is the protocol meaning?

Evaluation: in general, this is a phase in which quantitative indexes are provided, derived by comparison with a ‘truth’. By example, the effectiveness of discovery is usually measured comparing what is discovered with what is discovered in a controlled bunch of (meta)data. I invite authors to offer such quantitative measures.

Notes on English

Lines 17-18: no principal verb in this sentence

Lines 38-39: on the other end, this sentence has two principal verbs

Line 66: “to be used”

Line 87: probably not “Two functionalities” but “The functionalities”

Reviewer 4 Report

  1. In order for the survey to be reliable and valuable, in the reviewer’s opinion it is necessary to conduct research on a larger number of users than eight.
  2. The layout of the Chapter 3. Results and interpretation could be more reader-friendly. Especially the way of presenting and discussing the answers may lead the reader to confusion. Please consider bullet points or other forms of presentation of the cited answers.
  3. The table 1 is unclear. What information does it provide? Is it necessary to use a table to provide this information?
  4. There are too many self-citations in the document. I understand that two Authors deal with the topics covered in this article and have extensive scientific achievements in this field, but six quotes seem definitely redundant.
  5. The manuscript needs heavy editing:
  • Some sentences are unclear or have strange or incomplete syntax, e.g. „Metadata systems including standards and user interfaces designed by spatial data custodians to manage spatial data and make data sharing, discovery and use possible for the end users.”
  • There is no chapter 4, but two chapters 6 are present.
  • Some elements of the template document appear in the document although they are not completed and are not needed. See Chapter 6. Patents, Supplementary Materials, and Author Contribution.
  • Citation is incorrect. Cited articles should appear in the References in order of occurrence in the text.